# Impact of the First Wave of COVID-19 on Pediatric Oncology and Hematology: A Report from the French Society of Pediatric Oncology

**DOI:** 10.3390/cancers12113398

**Published:** 2020-11-17

**Authors:** Jérémie Rouger-Gaudichon, Eric Thébault, Arthur Félix, Aurélie Phulpin, Catherine Paillard, Aurélia Alimi, Benoît Brethon, Elodie Gouache, Sandra Raimbault, Eva de Berranger, Marilyne Poirée, Séverine Bouttefroy, Nicolas André, Virginie Gandemer

**Affiliations:** 1Department of Pediatric Oncology and Hematology, University Hospital of Caen (CHU Caen), 14000 Caen, France; 2Department of Oncology for Child and Adolescent, Gustave Roussy, 94800 Villejuif, France; Eric.THEBAULT@gustaveroussy.fr (E.T.); Arthur.FELIX@gustaveroussy.fr (A.F.); 3Department of Pediatric Oncology and Hematology, University Hospital of Nancy (CHRU Nancy), 54000 Nancy, France; a.phulpin@chru-nancy.fr; 4Department of Pediatric Hematology and Oncology, Hôpital Hautepierre, 67200 Strasbourg, France; catherine.paillard@chru-strasbourg.fr; 5Department of Pediatric, Adolescent, Young Adult, Institut Curie, CEDEX 05, 75248 Paris, France; aurelia.alimi@curie.fr; 6Department of Pediatric Hematology, Robert Debre Hospital, APHP, 75019 Paris, France; benoit.brethon@aphp.fr; 7Department of Pediatric Hemato-Oncology, Hospital Armand Trousseau, 75012 Paris, France; elodie.gouache@aphp.fr; 8Pediatric Oncology Unit, Centre Oscar Lambret, 59000 Lille, France; sandra.raimbault@gmail.com; 9Department of Pediatric Hematology, University Hospital of Lille (CHU Lille), 59000 Lille, France; Eva.DEBERRANGER@chru-lille.fr; 10Department of Pediatric Oncology and Hematology, University Hospital of Nice, 06000 Nice, France; poiree.m@chu-nice.fr; 11Institute of Pediatric Hematology and Oncology IHOPE, Centre Leon Berard, Hospices Civils de Lyon, 69002 Lyon, France; Severine.BOUTTEFROY@lyon.unicancer.fr; 12Department of Pediatric Hematology and Oncology, Hôpital pour enfant de La Timone, AP-HM, 13005 Marseille, France; nicolas.andre@ap-hm.fr; 13Department of Paediatric Hemato-Oncology, University Hospital of Rennes, University Rennes 1, 35000 Rennes, France; virginie.gandemer@chu-rennes.fr

**Keywords:** SARS-CoV-2, COVID-19, child, neoplasm

## Abstract

**Simple Summary:**

Data regarding coronavirus disease 2019 (COVID-19) description are still limited in pediatric oncology. The French society of pediatric oncology (SFCE) initiated a study to better describe COVID-19 presentation and evolution in patients followed in French pediatric oncology and hematology wards. By describing COVID-19 in this specific population, we aimed to identify the patients who may be the most at risk of severe COVID-19 and establish specific recommendations.

**Abstract:**

Data regarding coronavirus disease 2019 (COVID-19) description are still limited in pediatric oncology. The French society of pediatric oncology (SFCE) initiated a study to better describe COVID-19 in patients followed in French pediatric oncology and hematology wards. All patients diagnosed with COVID-19 and followed in a SFCE center were enrolled. Data from medical records were analyzed for all patients enrolled up to the end of May 2020. Data were available for 37 patients. Thirty-one were children under 18 years of age. Nineteen patients were female. Seventeen patients had a solid tumor, 16 had a hematological malignancy and four recently underwent hematopoietic stem cell transplantation (HSCT) for non-oncological conditions. Twenty-eight patients presented symptoms, most often with fever, cough, rhinorrhea and asthenia. Ground-glass opacities were the most frequent radiological finding with abnormalities mostly bilateral and peripherally distributed. Twenty-four patients received chemotherapy a month prior to COVID-19 diagnosis. Most patients did not require hospitalization. Three patients required oxygen at the time of diagnosis. In total, five patients were admitted in an intensive care unit because of COVID-19 and one died from the disease. Children and young adults treated for a cancer and/or with a HSCT may be at risk for severe COVID-19 and should be closely monitored.

## 1. Introduction

Coronavirus disease 2019 (COVID-19), resulting from severe acute respiratory syndrome coronavirus 2 (SARS-CoV-2) appears to affect children less severely than adults, with majority of benign forms and asymptomatic cases [1]. High mortality rates have been recently reported in adult patients with cancer [2,3]. Data are still limited in children with cancer while some reports suggested that they may be quite safe regarding the infection [4,5,6]. Although recommendations were initially made by the major child cancer organizations, new challenges have emerged [7,8]. A better description of the disease and its associated clinical and logistical challenges is available. As a result, pediatric oncology departments around the world can develop appropriate strategies [9].

To characterize clinical presentation and outcomes of children, adolescent and young adults with cancer and infected by SARS-CoV-2, the French society of pediatric oncology (Société Française de lutte contre les Cancers et leucémies de l’Enfant et de l’adolescent—SFCE) initiated a study (PEDONCOVID) to retrospectively and prospectively collect data regarding COVID-19 in this specific population. France has been so far one of the most affected countries with currently a total of 188,450 cases and 28,943 deaths [10]. We describe here the detailed data of 37 patients enrolled in French pediatric oncology centers from the SFCE.

## 2. Results

As of the 28th of May 2020, 41 patients with COVID-19 have been declared in SFCE centers. Most of patients were from Eastern France and the region around Paris, consistent with the distribution of the epidemic in whole population (Figure 1). Thirty-seven patients were included in the study. The main characteristics of patients are shown in Table 1.

Of the 37 patients, 31 were children under the age of 18 (mean: 11.2 years (1–25)). Nineteen patients (51%) were female. Seventeen patients had a solid tumor, 16 had a hematological malignancy and four recently underwent HSCT for non-oncological conditions. Nine patients were treated for a cancer relapse. Contact with an infected person was reported for 19 patients, with a median estimated incubation period of 9.5 days (data available for 10 patients). Twenty-eight patients presented symptoms (Table 2).

All patients were tested for SARS CoV-2 infection by PCR on a nasopharyngeal swab, which was positive in 34 patients (92%). Two patients were diagnosed with positive specific IgM serology and one patient was diagnosed upon typical clinical and radiological findings. Thoracic computed tomography-scan was performed in fifteen patients and was abnormal in all but one case. Ground-glass opacities were the most frequent abnormality (9 cases). Abnormalities were mostly bilateral and peripherally distributed. Blood count results were available for 31 patients. Lymphopenia below 0.5 G/L was found in thirteen patients. Neutropenia below 0.5 G/L was found in thirteen patients. Six patients presented concomitant neutropenia and lymphopenia. C-reactive protein (CRP) dosage results were available for 24 patients. Fifteen patients had an elevated CRP, with a CRP mean level of 41.6 mg/L (15–280). Four patients had a CRP above 50 mg/L. Liver enzymes were measured in seventeen patients, nine of whom had elevations greater than twice the upper limit of normal. A co-pathogen was found in blood samples of two patients (one with Staphyloccocus Epidermidis, and one with Epstein-Barr virus). Twenty-four patients received chemotherapy, on average 12.6 days (0–35) prior to COVID-19 diagnosis, with a mean number of drugs of 2.1 (1–5). Nine patients were currently (or recently) treated with less than 1 mg/kg/day of corticosteroids, five with cyclosporin A and one with tacrolimus. Overall, 20 patients were hospitalized at diagnosis but most of them did not require hospitalization specifically to manage COVID-19. There were significant discrepancies among pediatric oncology and hematology wards regarding schedule triage, rules of admission to hospital and SARS-CoV-2 testing. Seventeen patients received antibiotics, with ten patients presenting febrile neutropenia. Only one patient received remdesivir and two received hydroxychloroquine outside of a clinical trial.

In total, five patients were admitted into an ICU because of COVID-19 and one died from the disease. Their main characteristics are described in Table 3. Three patients required oxygen at the time of diagnosis and two of them were transferred into an ICU. The patient who received remdesivir had a long stay in ICU with neurological complications and SARS-CoV-2 was still detectable in nasopharyngeal swab and stool 20 and 28 days after COVID-19 diagnosis, respectively. Two injections of tocilizumab were performed as well. One patient who received hydroxychloroquine was discharged from ICU after a 10 days-long stay and was free of symptoms. SARS-CoV-2 test was not repeated for this patient. The other patient who received hydroxychloroquine presented COVID-related complications did not eliminate the virus and died from the infection despite the injection of two doses of tocilizumab.

Notably, four patients had been heavily treated for their cancer and/or recently underwent HSCT. Except for these patients, there were only asymptomatic to moderate forms of infection, with a median follow-up of 21 days (0–58). For the 13 patients for whom PCR had been repeated and these data were available, the mean time to a negative PCR was 16.5 days (7–28). COVID-19 delayed oncology treatment in 16 patients for an average of 14 days.

## 3. Discussion

The objective of this study was to describe the clinical presentation of COVID-19 in the pediatric oncology and hematology population, not to determine the incidence of COVID-19. Indeed, each center may have different screening strategies and the data collected in this study do not allow for the calculation of the incidence of COVID-19. Our data show that clinical and radiological descriptions of COVID-19 are quite similar to those reported in adults, and that incubation period or virus clearance appear to be the same in this population as what has been described before. Most of patients developed a mild or even an asymptomatic form of the disease. However, as we previously stated, [11] and in contrast with other reports [4,5,6], we found that patients may be at a higher risk of developing a severe form of COVID-19. Indeed, five patients (15%) required admission into an intensive care unit (ICU) and one patient died from COVID-19 complications. Among patients with a cancer history, one had high-grade glioma, the others had a relapsed acute lymphoblastic leukemia (ALL) and were highly immunocompromised. The deceased patient was a four-year-old boy undergoing induction chemotherapy for relapsed ALL. His respiratory state worsened, and he was admitted in an ICU, where he developed intense macrophage activation syndrome and subsequent complications that led to his death. In the study from Boulad et al., only one patient required a noncritical care hospitalization and the COVID-19 manifestations were mild for most patients [4]. This is concordant with the data from Spain where no severe case was reported [5]. In contrast, in New York, the joint experience form the memorial Sloan Kettering Cancer Center and New York Presbyterian Hospital shows that among 19 patients infected with SARS-CoV-2, five (20%) required intensive care including mechanical ventilation and a 12-year-old boy with hemoglobinopathy developed acute chest syndrome and died from COVID-19 complications. In Lombardia, Ferrari and al. reported COVID-19 complications in 2/21 patients [12]. We have no clear explanation for such differences between our work and these reports and it may be only due to happenstance. Indeed, ICU admission criteria are likely similar in France and other Western countries and there is no reason that French patients may be more immunocompromised or vulnerable than their US, Spanish or Italian counterparts. In adults, recent reports suggest that patients currently treated for a cancer or with a history of cancer more frequently develop severe COVID-19 compared to the general population [2,13,14]. Only three patients received an anti-SARS-CoV-2 treatment. Because of this small number of patients, outcomes are difficult to appreciate. Tolerance of the administrated treatments was not specifically evaluated in this study, but no complications were reported. Twenty patients were hospitalized, but the rules of admission were not uniform among centers and the reasons for hospitalization were not always specified. Thus, any interpretation of this high number of hospitalizations must be taken very carefully.

Oncologic treatment was delayed in almost half of cases, which indicates that COVID-19 impacted the care of patients even if most cases were mild. Notably, some patients had their treatment as planned and did not develop any complications. However, every situation is unique and clinicians who followed these patients made their decision regarding the patients’ own history and potent risk factors, so that no general recommendation to pursue oncologic treatment may be stated from this study. On the contrary, delaying a non-urgent oncologic treatment appears to be wise in the context of suspected or proven infection, as suggested by most clinicians [15]. Conversely, delaying the initial management of children with cancer may be dramatic. Thus, in Phildelphia, Ding et al. reported five patients who became critically hill because of delay in diagnosis illustrating indirect impact on morbidity of COVID-19 [7].

A striking difference between our data and the first published reports is the sex ratio [4,5]. Though there were slightly more females than males in our study, our cohort is equilibrated regarding sex distribution. In some reports concerning pediatric oncology populations, there was a strong proportion of males with 5 to 15 times more male than female [4,5]. Similarly, Gampel and al. reported that among the five patients requiring intensive care, all were males [16]. Other studies in pediatric general population [17] or adult oncology population [2] did not find this high proportion of males. More recently, Bisogno et al. described COVID-19 management in 29 children treated by chemotherapy and/or immunotherapy. The sex ratio was comparable to those from our study, with 16 females and 13 males [18]. Thus, we cannot explain the difference between our study and the work from Boulad et al. and de Rojas et al. but there is currently no clue for a specific sex distribution among children and/or oncology patients.

Biological findings are difficult to interpret since a high proportion of patients recently received drugs that could alter blood tests. However, profound neutropenia and profound lymphopenia were found in almost one third of patients. Even if that can be explained by the oncologic treatment received, the infection likely participated and increased the cytopenias. Lymphopenia is common with SARS-CoV-2 infection, and it appears that the deeper the lymphopenia, the more severe the disease [19,20]. Our cohort is too small to address this question even if two of the five patients admitted into an ICU had profound lymphopenia. For the same reason, it is not possible to draw a conclusion on the association between PCR level and disease severity. However, it should be noted that CRP level was quite low in our cohort and that of the four patients with elevated CRP level above 50 mg/L, two were admitted to an ICU.

Clinicians should pay attention to lung lesions caused by COVID-19 for patients who may receive treatments that could themselves damage lung tissue. Indeed, delaying lung irradiation or administration of drugs with known lung toxicity may be considered for infected patients. However, this delay should not be too long to avoid the risk of progression. In our cohort, the mean delay in treatment administration was 14 days, which appears reasonable regarding both this risk of disease progression and the risk of complications related to COVID-19. Of course, making the decision of delaying treatment and the length of this delay should be considered carefully and should depend on the situation.

Until we get more data, we advocate not reassuring parents regarding the mild forms of COVID-19 in patients treated for malignancies. We recommend that patients with severe hematological malignancies and/or receiving heavily immunosuppressive treatments should be carefully watched and protected from the COVID-19 risk. This shall be of interest for patients in South Asia and Latin America who are now facing the COVID-19 pandemic and countries facing the second wave.

## 4. Materials and Methods

All patients followed in a SFCE center and diagnosed with a SARS-CoV-2 infection who had undergone anti-cancer treatment in the past 6 months were enrolled in the study. We also included all patients from SFCE centers who underwent a hematopoietic stem-cell transplantation (HSCT) for any reason and with immunosuppressive therapy either ongoing or interrupted less than 6 months prior. Since SFCE centers may provide care to children and adolescents as well as in young adults, there was no limit of age to enroll patients in the study.

Diagnosis criteria for SARS-CoV-2 infection were as follows: (1) biologically proven infection with positive polymerase chain reaction (PCR) or positive specific positive IgM SARS-CoV-2 serology, (2) clinical and radiological diagnosis with at least two (if any contact with a suspected or confirmed COVID-19 case) or three (if no notion of contact) clinical signs (comprising fever, cough, loss of smell, loss of taste, myalgia, chest pain, dyspnea, respiratory distress syndrome, rhinorrhea, diarrhea, headaches, asthenia and skin rash) and computed-tomography (CT)-scan signs (ground glass opacities, crazy-paving, linear condensations with a peripheral and/or bilateral distribution) compatible with SARS-CoV-2 infection. All patients and their families were informed of this study. The study was approved by the ethics committee of the coordinating center (N04/2020/ROU). This study is registered in clinicaltrial (NCT04433871).

## 5. Conclusions

We deliver here the first description of the French pediatric oncology and hematology cohort of patients diagnosed with SARS-CoV-2 infection, which to our knowledge is the largest ever published to date. Consistent with previous reports [4,5], relatively few pediatric cancer patients had clinical signs of COVID-19 or tested positive for the virus. Nevertheless, as previously noted [11], some highly immunocompromised patients are at risk of developing severe forms of COVID-19, notably those with hematological malignancies, who have been heavily pretreated and/or with a history of HSCT, which justifies social distancing and specific triage before admission in oncology units.

## Figures and Tables

**Figure 1 cancers-12-03398-f001:**
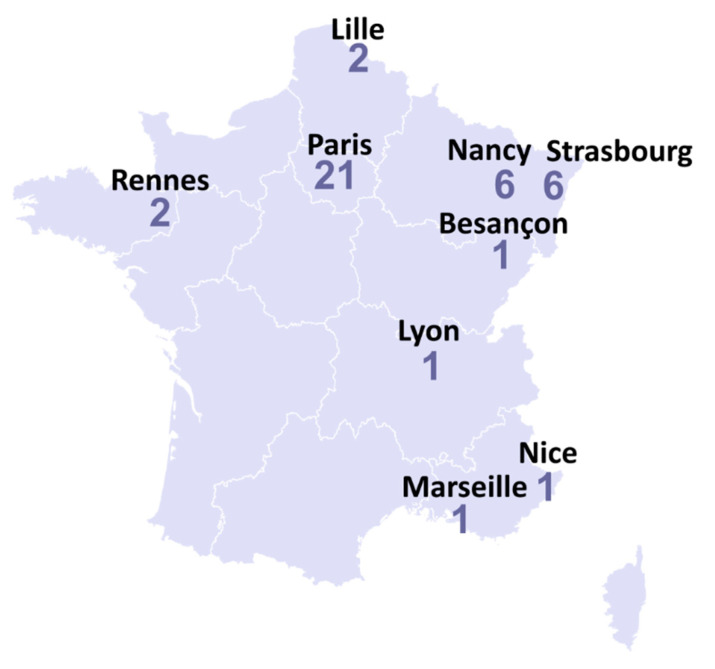
Reported COVID-19 cases in French pediatric oncology and hematology wards up to the 28th of May.

**Table 1 cancers-12-03398-t001:** Main characteristics of enrolled patients.

Characteristics	Number of Patients	Percentage
Sex		
- Male	18	49
- Female	19	51
Age		
- 0–2 years	3	8
- 2–6 years	8	22
- 6–12 years	9	24
- 12–18 years	11	30
- Above 18 years	6	16
Pathological condition		
Hematological malignancy	16	43
- ALL	10	27
○ t (9;22) negative	9	24
○ t (9;22) positive	1	3
- AML	1	3
- CML	1	3
- NHL	4	11
Solid tumor	17	46
- Localized	11	30
- Metastatic	6	16
- CNS tumor	7	19
- Medulloblastoma	2	5
- CNS NB-FOXR2	1	3
- Pinealoblastoma	1	3
- High-grade glioma	1	3
- Low-grade glioma	1	3
- Cerebral ATRT	1	3
- Bone tumor	4	11
- Osteosarcoma	2	5
- Ewing sarcoma	2	5
- MPNST	1	3
- Wilms’ tumor	1	3
- Neuroblastoma	1	3
- ATRT of the kidney	1	3
Non-oncologic condition	4	11
- Aplastic anemia	1	3
- Sickle cell disease	1	3
- EBV-induced macrophage activation syndrome	1	3
- Familial septic granulomatosis	1	3
COVID-19 diagnosis method		
- Positive SARS-CoV-2 PCR	34	92
- Positive serology	2	5
- Clinical and radiological diagnosis	1	3
Symptoms		
- Yes	28	76
- No	9	24
Treatments one month prior to COVID-19 diagnosis		
- Chemotherapy	24	65
- Corticosteroids	9	24
- G-CSF	7	19
- Immunosupressive agents	6	16
- Targeted therapy or moncolonal antibody	5	14
- Radiotherapy	2	5
- Surgery	2	5
- None	1	3

Abbreviations: ALL: acute lymphoblastic leukemia; AML: acute myeloid leukemia; ATRT: atypical teratoid rhabdoid tumor; CML: chronic myeloid leukemia; CNS: central nervous system; CNS NB-FOXR2: CNS neuroblastoma with FOXR2 activation; COVID-19: coronavirus disease 2019; EBV: Epstein-Barr virus; G-CSF: granulocyte-colony stimulating factor; MPNST: malignant peripheral nerve sheath tumor; NHL: non-Hodgkin lymphoma; SARS-CoV-2: severe acute respiratory syndrome coronavirus 2.

**Table 2 cancers-12-03398-t002:** Clinical signs of COVID-19 among the 28 symptomatic cases.

Symptoms	Number of Patients	Percentage of Symptomatic Cases
Fever	20	71%
>38.5 °C	14	50%
Between 38 °C and 38.5 °C	6	21%
Cough	14	50%
Rhinorrhea	12	43%
Asthenia	12	43%
Loss of smell/taste	8	29%
Diarrhea	7	25%
Chest pain	6	21%
Myalgia	5	18%
Respiratory distress signs	5	18%
Tachycardia	4	14%
Headaches	3	11%
Skin rash	2	7%
Neurological signs	2	7%

**Table 3 cancers-12-03398-t003:** Clinical characteristics of patients admitted in an intensive care unit for COVID-19.

Age (Years)	Sex	Pathology	Time to ICU Admission (days)	Type of Respiratory Support	Specific Treatment Against SARS-CoV-2	Evolution	ICU Stay (Days)	Biology	Comments
12	M	Relapsed B-ALL, HSCT 2 months prior SARS-CoV-2 infection, aGVHD	1	Non-invasive ventilation	No	Favorable	5	Neutro:3.2 G/L Ly: 0.07 CRP 39 mg/L	Developed cerebral toxoplasmosis after ICU stay
5	F	SS sickle cell disease with cerebral vasculopathy, HSCT 1 month prior to SARS-CoV-2 infection	2	Mechanic ventilation for 5 days	Remdesivir Tocilizumab (2 injections)	Favorable	29	Neutro:5.12 G/L Ly: 1.28 CRP <10 mg/L Ferritin 4400 µg/L	Neurological complications (bilateral facial palsy, progressive acute polyneuropathy) SARS-CoV-2 still detectable in stool at day 28 of infection
8	F	Relapsed high-grade astrocytoma	10	Non-invasive ventilation	No	Favorable	Unknown	WBC: 2.7 G/L Neutro: 0.5 G/L CRP max: 251 mg/L	Repeat PCR negative at day 8
18	F	Relapsed B-ALL (2nd relapse), treated with vincristine only	4	Non-invasive ventilation	Hydroxychloroquine	Favorable	10	Neutro: 0.7 G/L Ly: 0.48 G/L CRP max: 267 mg/L	CAR-T cell treatment delayed because of SARS-CoV-2 infection
4	M	Relapsed B-ALL	5	Mechanic ventilation for 2 days	Hydroxychloroquine Tocilizumab (2 injections)	Deceased (19 days after COVID-19 diagnosis)	14	WBC: 0.12 G/L CRP <10 mg/L Ferritin > 300,000 µg/L	Severe macrophage activation syndrome

Abbreviations: aGVHD: acute graft versus host disease; ALL: acute lymphoblastic leukemia; CAR: chimeric antigen receptor; COVID-19: coronavirus disease 2019; HSCT: hematopoietic stem cell transplantation; ICU; intensive care unit; PCR: polymerase chain reaction; SARS-CoV-2: severe acute respiratory syndrome coronavirus 2.

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
