# Peer review of "Impact of the First Wave of COVID-19 on Pediatric Oncology and Hematology: A Report from the French Society of Pediatric Oncology"

_cancers, 2020, doi:10.3390/cancers12113398_

Round 1

Reviewer 1 Report

The topic is an interesting one.

The authors need to mention about the treatment offered to the study patients, their response to treatment and any complications as a result of the treatment offered.

Any measurement of plasma cytokines was done in these study population. If so, were they any different form those only with COVID-19 without cancer. Any changes in the plasma cytokines after treatment for cancer compared to those without cancer but had COVID-19 and treated with current modalities of therapy. 

Author Response

Thank you for your useful comments. Please find below our answers.

The authors need to mention about the treatment offered to the study patients, their response to treatment and any complications as a result of the treatment offered.

We added a paragraph in the results section to better describe specific anti-SARS-CoV-2 treatments that some patients received. We also added a paragraph in the discussion to comment on that. Modifications appear in yellow in the main text. Actually, only three patients received such treatments and no one presented complications. However, outcomes are difficult to interpretate because of the small number of patients.

Any measurement of plasma cytokines was done in these study population. If so, were they any different form those only with COVID-19 without cancer. Any changes in the plasma cytokines after treatment for cancer compared to those without cancer but had COVID-19 and treated with current modalities of therapy. 

Cytokines were not monitored in this study, and such measurement were not reported, so we cannot comment about that and we cannot compare with another cancer-free population.

Reviewer 2 Report

Minor revisions

Material and methods paragraph should be shifted after the introduction paragraph.

Regarding the tables, no legenda for the abbreviations are reported, please include a legenda below each table.

Table 2 reported the clinical sign of COVID-19 for 26 symptomatic cases but in the text Authors reported that the patients presenting symptoms were 28. Please check and correct.

Authors reported that “most patients did not require hospitalization..”. Please report the number of patients that were hospitalized included the 5 patients admitted to PICU.

No information about the way to admit patients to hospital, triage schedule or the rule for containment of infection were described.

In discussion, Authors reported that in their paper, in contrast to other reported paper by the USA and Spain colleagues, the sex ratio is slightly in favour of female. Actually, the paper by Bisogno G, in J Pediatric Infect Dis Soc 2020, a study from the Infectious Diseases Working Group of the AIEOP, reported a cohort of 16 females and 13 males. Please add this reference and modify discussion accordingly.

Typos must be checked. In particular, in discussion, line 154 and conclusion, line 201.

Author Response

We would like to thank you for your constructive comments that will improve our manuscript. We consequently modified our manuscript (modifications appear in green in the text). As asked, we will answer point by point:

"Material and methods paragraph should be shifted after the introduction paragraph."

We totally agree with this comment, it would be easier to understand by doing this shift. However, we have prepared this manuscript as asked by the editor.

"Regarding the tables, no legenda for the abbreviations are reported, please include a legenda below each table."

You are right, we made the correction.

"Table 2 reported the clinical sign of COVID-19 for 26 symptomatic cases but in the text Authors reported that the patients presenting symptoms were 28. Please check and correct."

Thank you for noticing this mistake. We corrected it in the table.

"Authors reported that “most patients did not require hospitalization..”. Please report the number of patients that were hospitalized included the 5 patients admitted to PICU.

No information about the way to admit patients to hospital, triage schedule or the rule for containment of infection were described."

We did not specify the numbers initially because of the potential heterogeneity of hospitalization indications according pediatric oncology wards. However, we took into account your remark and modified the manuscript to give more detail.

"In discussion, Authors reported that in their paper, in contrast to other reported paper by the USA and Spain colleagues, the sex ratio is slightly in favour of female. Actually, the paper by Bisogno G, in J Pediatric Infect Dis Soc 2020, a study from the Infectious Diseases Working Group of the AIEOP, reported a cohort of 16 females and 13 males. Please add this reference and modify discussion accordingly."

We did not noticed that publication at the time of writing. Thank you for this suggestion that we took into account.

"Typos must be checked. In particular, in discussion, line 154 and conclusion, line 201"

We corrected these mistakes. Thank you.

Reviewer 3 Report

  1. Please put the items at the left hand side in “Table 1. Main characteristics of enrolled patients.” and “Table 2. Clinical signs of COVID-19 among the 26 symptomatic cases.”.
  2. 145: On the contrary, delaying a non-urgent oncologic treatment appears to be wise in the context of suspected or proven infection, as suggested by most clinicians (in press). Oppositely, delaying the initial management of children with cancer may be dramatic. P.172: In our cohort, the mean delay intreatment administration was 14 days, which appears reasonable regarding both this risk of disease progression and the risk of complications related to COVID-19.
    • Did the suggestion be suitable for any kind of Pediatric Oncology?
  3. Author seemed to discuss gender and age effect, did race, SES cause any different?

Author Response

Thank you for your helpful comments. Here are our answers:

Please put the items at the left hand side in “Table 1. Main characteristics of enrolled patients.” and “Table 2. Clinical signs of COVID-19 among the 26 symptomatic cases.”.

Modifications have been made accordingly.

145: On the contrary, delaying a non-urgent oncologic treatment appears to be wise in the context of suspected or proven infection, as suggested by most clinicians (in press). Oppositely, delaying the initial management of children with cancer may be dramatic. P.172: In our cohort, the mean delay intreatment administration was 14 days, which appears reasonable regarding both this risk of disease progression and the risk of complications related to COVID-19.

    • Did the suggestion be suitable for any kind of Pediatric Oncology?

Thank you for your pertinent remark. We realize that the message may be  unclear. We added a sentence at the end of the paragraph to to clarify our thoughts (modifications in red).

Author seemed to discuss gender and age effect, did race, SES cause any different?

It would had been interesting to study these specific points. Unfortunately, such data were not collected in this study. Besides, data about race cannot be recorded according to the France legislation.

Round 2

Reviewer 1 Report

nil.